# Synthesis, Crystal Packing Aspects and Pseudosymmetry in Coordination Compounds with a Phosphorylamide Ligand

**Taisiya S. Sukhikh ***[ID]**, Radmir M. Khisamov and Sergey N. Konchenko**

Nikolaev Institute of Inorganic Chemistry, Siberian Branch of the Russian Academy of Sciences, 3 Lavrentiev Ave., 630090 Novosibirsk, Russia
*   Correspondence: sukhikh@niic.nsc.ru

**Abstract:** This work reports the synthesis and crystal structure of new closely related coordination compounds, $[ML_2]\cdot n$THF, where M is Zn or Mn; L is a phosphorylmethylamide derivative of benzothiadiazole; $n = 1$ (M = Zn) and 1, 2 (M = Mn); and THF is tetrahydrofuran. The zinc compound, **1**·THF, crystallizes in a high-symmetry space group, $I4_1/a$, that is relatively rare for compounds with organic ligands. The corresponding manganese congener, **2**·THF, with a similar crystal packing, features a pseudosymmetrical structure $P2_1/c$ of the doubled volume of the unit cell as compared to **1**·THF. The main difference between the structures lies in a different arrangement of solvate THF molecules, which likely modulates the crystal packing of the complexes. Another manganese solvatomorph, **2**·2THF, reveals a fundamentally different crystal packing while exhibiting a similar geometry of the complex. We consider the problem of localization of solvate THF molecules and the types of their disorder by the example of compounds **1**–**2**.

**Keywords:** crystal structure; coordination compounds; pseudosymmetry; solvates





## 1. Introduction

The synthesis of novel heterocyclic compounds and related coordination compounds is a hot topic of steadily growing interest. The rationale for the design of such compounds covers, but is not limited to, areas of sensors [1–3], solar energy converters [4,5], catalysts [6], energy storage and energy transfer devices [7–10], and pharmaceutics [11–13]. For these applications, the solid-state structure is an important characteristic in determining the final properties of compounds [14–18]. In particular, the study of the structure of (pseudo)polymorphs attracts attention, which is also of fundamental interest, prompted by the question of which forces/conditions govern one or another crystal packing [19–21]. Molecules or ions, which generally have a complicated shape with a number of convexities and hollows, can engage in different types of junctions with each other (not necessarily through specific interactions), thus conforming different polymorphic modifications of heterocyclic compounds [22–24]. Another interesting feature stemming from the complicated shape of the molecules is their ability to form different solvates (solvatomorphs): solvate molecules can modulate the junctions between host molecules, resulting in a wide variety of crystal packing patterns [25–27]. Organic solvent molecules typically play the role of space fillers in a crystal [28]. In the absence of strong specific interactions with a host, a solvate molecule can exhibit disorder whose nature depends on the shape of the cavity that accommodates the molecule [29]. More specifically, the disorder arises in cases where different, approximately thermodynamically equivalent variants of the arrangement of a molecule exist. Some minor changes in crystallization conditions or phase transitions can induce the rearrangement of molecules from a symmetrical disordered case to an ordered pseudosymmetric one. Pseudosymmetry is an interesting but occasionally overlooked feature of some crystals [30], which arises as a result of the violation of a crystal symmetry by some of their constituting fragments [31–35], e.g., solvent molecules.

In this work, we report the synthesis and crystal structure of new compounds, [ML$_2$]·$n$THF, where M = Zn, Mn; L = benzo[c][1,2,5]thiadiazol-4-yl((diphenylphosphoryl)(phenyl)methyl)-amide; $n$ = 1 (M = Zn) and 1, 2 (M = Mn); and THF = tetrahydrofuran. The complexes [ML$_2$] (hereafter, a molecule of a complex will be referred to as a complex) of approximately the same geometry co-crystallize with solvent THF molecules, resulting in different solvatomorphs. We discuss crystal packing aspects and pseudosymmetry in the crystals.

## 2. Materials and Methods

### 2.1. General

All manipulations for the syntheses were performed using the Schlenk technique and a glovebox. Solvents were purified using the standard technique and stored under an argon atmosphere. HL was synthesized as described recently [36]. Elemental analyses were performed on various MICRO cube instruments (Langenselbold, Germany) for C, H, N, and S elements. IR spectra were recorded on a Fourier IR spectrometer FT-801 (Simex; Novosibirsk, Russia) in KBr pellets (Figure S1). $^1$H NMR spectra (500.13 MHz) and $^{31}$P{H} NMR spectra (202.45 MHz) were obtained with a Bruker DRX-500 spectrometer (Madison, WI, USA) in dry C$_6$D$_6$ (Figures S2–S4); the solvent peak was used as an internal reference.

### 2.2. X-ray Data

Single-crystal XRD data for **1**·THF were collected at 150 K with a Bruker X8 diffractometer (Madison, WI, USA) with a CCD detector and graphite-monochromated source (MoK$_\alpha$ radiation). The data for compounds **2**·THF and **2**·2THF (Table S1) were collected at 150 K with a Bruker D8 Venture diffractometer with a CMOS PHOTON III detector (Bruker, Madison, WI, USA) and IµS 3.0 microfocus source (MoK$_\alpha$ radiation ($\lambda$ = 0.71073 Å), collimating Montel mirrors; Incoatec GmbH, Geesthacht, Germany). The crystal structures were solved using the SHELXT [37] and were refined using the SHELXL [38] programs with OLEX2 GUI [39]. Atomic displacement parameters for non-hydrogen atoms were refined in anisotropic approximation with the exception of the disordered THF molecules. For the latter, EADP constraints and SADI or DFIX restraints were applied where needed. Hydrogen atoms were placed geometrically and refined in the riding model. The volume of voids in the absence of the THF molecules was calculated without further structure refinement by means of the solvent-accessible voids procedure implemented in Olex2 with the same parameters for all the structures (the radius of probe sphere of 1.3 Å; the grid step of 0.2 Å). The structures of **1**–**2** were deposited to the Cambridge Crystallographic Data Centre (CCDC) as a supplementary publication, No. 2225583–2225585.

### 2.3. Syntheses

[ZnL$_2$]·THF (**1**·THF)

To a mixture of HL (100.0 mg, 0.226 mmol), potassium trimethylsililamide (KHMDS; 45.2 mg, 0.226 mmol) and ZnCl$_2$ (15.4 mg, 0.113 mmol) THF (5 mL) was added. The mixture readily turned to dark blue; it was stirred for a day and concentrated under vacuum. Violet crystalline product was formed. The solid was centrifuged and washed with THF (1 mL) and n-hexane (2 mL). The yield was 0.109 g (95%). Calc. for C$_{50}$H$_{38}$N$_6$O$_2$P$_2$S$_2$Zn·C$_4$H$_8$O (1018.45): C 63.7, H 4.6, N 8.2, S 6.3. Found C 63.6, H 5.0, N 7.3, S 6.0. $^{31}$P{H} NMR (C$_6$D$_6$, $\delta$, ppm): 40.20 (s). $^1$H NMR (C$_6$D$_6$, $\delta$, ppm): 8.11 (d, 4H, J = 7.7 Hz); 7.57 (dd, 4H, J = 10.5 Hz, 7.5 Hz); 7.45 (dd, 4H, J = 11.5 Hz, 7.0 Hz); 7.29 (t, 2H, J = 8.0 Hz); 7.15–7.10 (m, 6H); 6.98 (d, 2H, J = 3.0 Hz); 6.93 (t, 4H, J = 7.5 Hz); 6.86 (d, 2H, J = 7.0 Hz); 6.83–6.75 (m, 6H); 6.35 (d, 2H, J = 7.7 Hz); 5.85 (d, 2H, J = 7.6 Hz); 3.57 (m, 2H, THF), 1.41 (m, 2H, THF).

[MnL$_2$]·nTHF (**2**·nTHF)

To a mixture of HL (50.0 mg, 0.113 mmol), MnCl$_2$ (7.1 mg, 0.057 mmol) and KHDMS (22.6 mg, 0.113 mmol) THF (5 mL) was added. The mixture readily turned to dark blue; it was refluxed for a week. During this time, the white solid MnCl$_2$ disappeared, and dark violet crystalline product of **2**·THF was formed. The solution was concentrated twice under vacuum; the solid was separated and washed with THF (2 mL) and diethyl ether

(2 mL). The yield of **2**·THF was 0.497 g (87%). Calc. for $C_{50}H_{38}MnN_6O_2P_2S_2$·THF (1008.0): C 64.3, H 4.6, N 8.3, S 6.4. Found C 64.3, H 5.1, N 8.0, S 6.5. The $^{31}P\{H\}$ NMR spectrum of the compound dissolved in $C_6D_6$ showed only weak signals from traces of free HL. The absence of signals of the title compound in the selected range (Figure S4) is likely caused by a strong paramagnetic shift and broadening.

Compound **2**·2THF as dark violet crystalline solid was formed upon extraction of **2**·THF with THF in a two-section ampoule. A few crystals of the compound were formed, which was characterized only by means of single-crystal XRD.

## 3. Results and Discussion

### 3.1. Synthesis of the Compounds

Reactions between $MCl_2$ (M = Zn and Mn) and the proligand HL in the presence of potassium trimethylsililamide (KHDMS) as a base in THF afforded the corresponding complexes $[ML_2]$ (**1** and **2**, correspondingly; Scheme 1). In the case of **1**, the reaction easily proceeded at room temperature, while in **2**, we refluxed the mixture to involve all the amount of relatively poorly soluble $MnCl_2$ in the reaction. The complexes were precipitated from the mother liquor as solvates **1**·THF and **2**·THF. The formation of the compounds with similar yet distinctive features of the crystal structures (see the discussion below) can be governed by the different temperatures of the reaction/crystallization or by some effects of the metal, e.g., seed effects of $MnCl_2$. The latter is known to form polynuclear moderately soluble species with THF of the general formula $[MnCl_2(THF)_x]$ [40], which could temporarily crystallize as intermediate products in the course of the reaction. Interestingly, the subsequent extraction of solid **2**·THF with a portion of THF afforded another crystalline solvate, **2**·2THF. This also implies that the minor changes in the crystallization conditions affect the formation of one or another crystalline phase.

**Scheme 1.** Synthesis of compounds **1–2**.

### 3.2. Crystal Structure of the Compounds

According to the single-crystal XRD analysis, compounds **1**·THF and **2**·THF exhibit a similar crystal packing (Figure 1a,b) and can be referred to as isotypic ones. **1**·THF crystallizes in the tetragonal space group $I4_1/a$ and reveals half of the complex in the crystallographically independent part (Z' = 0.5). Meantime, **2**·THF features the pseudosymmetric structure (Figure S5), crystallizing in the monoclinic space group $P2_1/c$ (Z' = 2) with the monoclinic angle of 90.0900(10)° and the doubled unit cell volume as compared to **1**·THF (in the primitive setting). A comparison of reciprocal space reconstructions for the compounds is presented in Figure S6.

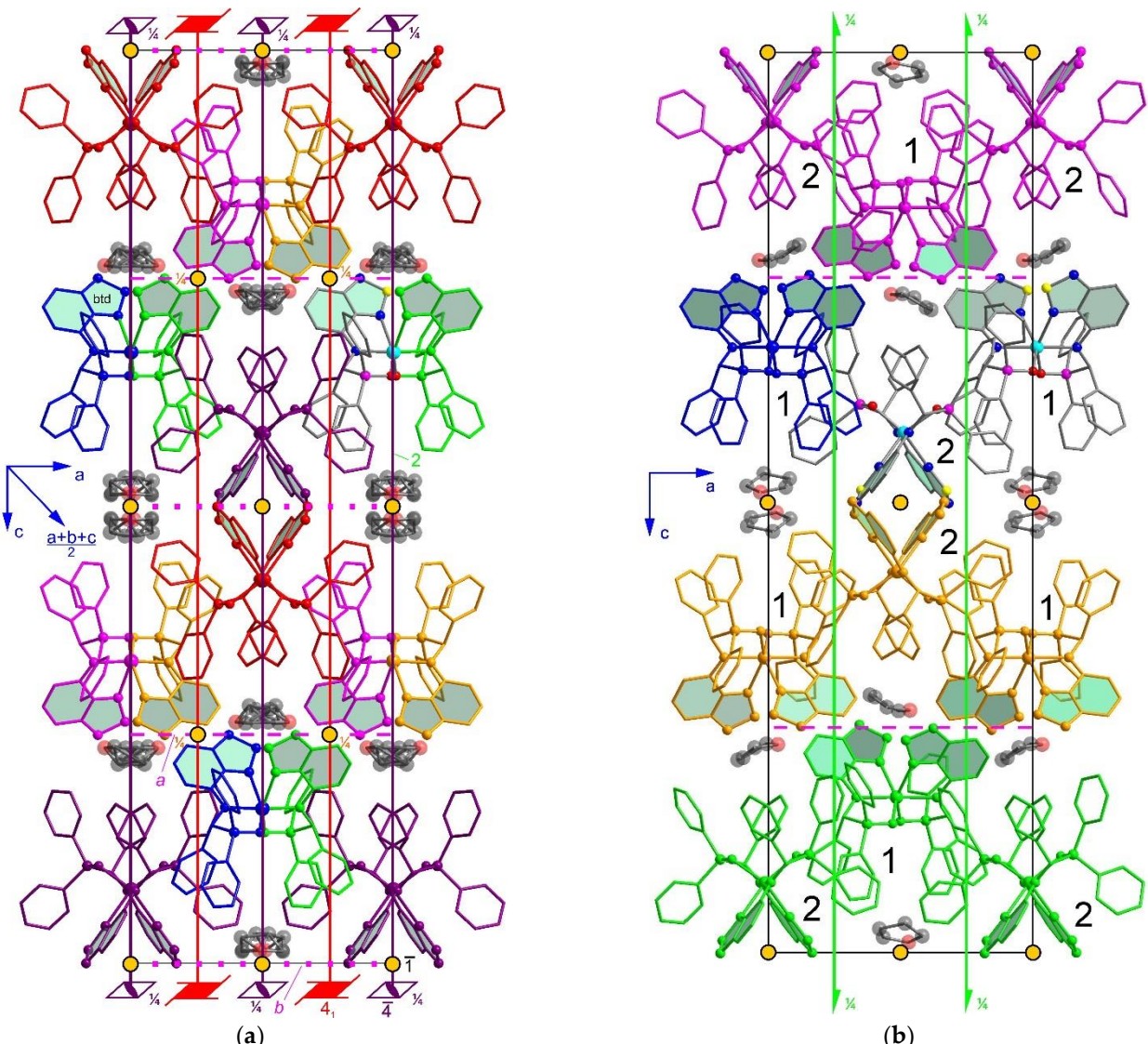

**Figure 1.** A fragment of crystal packing of compounds: (**a**) **1**·THF and (**b**) **2**·THF (in $P112_1/a$ setting; the numbers denote the first and the second independent molecule) showing the symmetry operations. Colour scheme: blue—translation, yellow—inversion, magenta—glide plane, green—2-fold axis (rotation or screw), red—$4_1$ axis, violet—(–4) axis. Crystallographically independent moiety is shown in grey, and symmetry-related moieties are coloured with the corresponding symmetry colour scheme. Hydrogen atoms are omitted and solvate THF molecules are shown as transparent spheres (without colours of symmetry relations).

Complexes **1** and **2** (Figure 2a) exhibit quite similar geometry. Two tridentate ligands coordinate with the metals in a meridional manner, forming an octahedral coordination environment $\{O_2N_4\}$ (Figure 2b). The continuous symmetry measures (CSM) analysis [41,42] gave similar values of deviation of the coordination polyhedra from the regular octahedron, being slightly larger for the case of **2** (Table S2). Donor atoms belonging to the same ligand are forced closer to each other than the corresponding positions in the regular octahedron due to the constringent effect of the specific geometry of the ligand. The {N–C–P–O} moiety, comprising the metallocycle in the complexes, is flattened as compared to the parent aminomethylphosphine HL (Table S2).

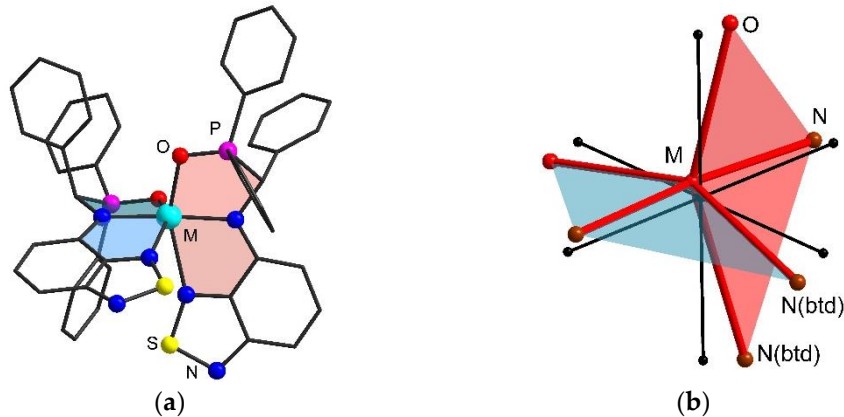

**Figure 2.** (**a**) The structure of complexes **1** and **2** exemplified by the former. (**b**) Representation of the atomic deviation of the coordination polyhedra from the regular octahedron. Metallocycles of two L ligands are coloured red or blue.

In terms of point symmetry, the complexes possess a local two-fold rotation axis, which in the case of **1**·THF coincides with the corresponding crystallographic axis, i.e., the Zn atom lies in the special position. In **2**·THF, the greatest atomic deviation from the pseudo-crystallographic symmetry is observed for the Ph groups (Figure S5). In the structures, two neighbouring complexes are rotated relative to each other by approximately 90°; this relation becomes strict in **1**·THF owing to the presence of the four-fold screw axis (Figure 1a). The main difference between the structures lies in a different arrangement of solvate THF molecules, which likely modulates the crystal packing of the complexes. Positions of the complexes in **1**·THF and **2**·THF are close to each other (Figure S7); their greatest inconsistency is observed in the positions of the phenyl groups arranged close to the solvate molecules. Thus, the geometry of complexes **1** and **2** is very similar, and the distortion degree of the latter from $C_2$ symmetry is low. Note that the structures with related Mn(II) complexes possessing $C_2$ symmetry have been reported elsewhere (e.g., with CCDC refcodes AQODAL [43], PAQQIL [44], and XEJWIW [45]). This suggests that there are no obvious restrictions for the formation of the tetragonal structure for **2**·THF. The Zn complex, in turn, could crystallize in the monoclinic space group.

Structure **1**·THF comprises one independent disordered THF, while in **2**·THF, two independent solvate molecules (Z′ = 2) are more ordered and show a displacement from the average "tetragonal" positions. Localization of the solvate molecules will be discussed below. Solvatomorph **2**·2THF reveals a crystal packing (Figure S8) that fundamentally differs from that of the previous compounds (space group $P2_1/c$, Z′ = 1) while exhibiting a similar geometry of the complex. The local two-fold rotation axes of the complexes in **2**·2THF are not parallel to the unique axis and arrange relative to the latter at an angle of 56°.

All the compounds reveal π–π interactions between the benzothiadiazole (btd) moieties of neighbouring molecules, which possess a similar relative arrangement owing to the similar molecular shape. In **1**·THF and **2**·THF, these btd–btd junctions with the local inversion symmetry compose complexes in chains spreading along two mutually perpendicular directions. Such π–π interactions with a head-to-tail arrangement of aromatic moieties are quite typical for nitrogen-containing heterocyclic derivatives [46]. In **2**·2THF, the junctions connect complexes in insular pairs. Outside the pair, the π–π interaction is prevented by the presence of one of the THF molecules between the btd moieties. Another crystallographically independent THF molecule in **2**·2THF and the molecule in **1**·THF and **2**·THF are arranged in a different manner with respect to the complexes. In **1**·THF and **2**·THF, the THF molecule is arranged above a $Ph_4$ portal of the complex composed of four phenyl groups of C(Ph) and P(Ph) moieties (Figure 3a). In **2**·2THF, the THF molecule is located outside the portal that is pointed to a portal of the neighbouring complex (Figure 3b).

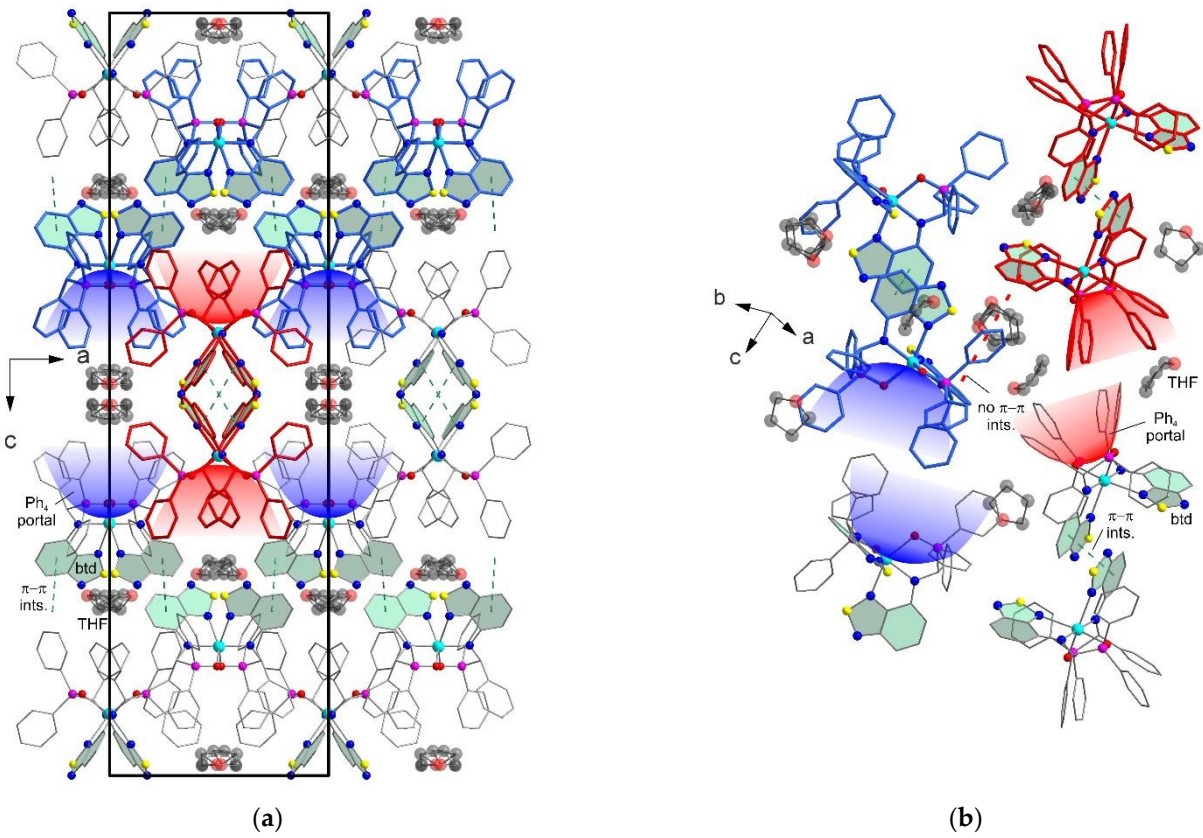

(**a**)　　　　　　　　　　　　　　　　　　　　　　　(**b**)

**Figure 3.** A fragment of crystal packing of compounds **1**·THF and **2**·THF exemplified by the former (**a**) and **2**·2THF (**b**) showing π–π interactions of btd moieties and relative arrangement of Ph$_4$ portals. Pairs of π–π interacting molecules are coloured blue or red. Hydrogen atoms are omitted and solvate THF molecules are shown as transparent spheres.

### 3.3. The Problem of Localization of Solvate Molecules

In the case of compound **1**·THF, we cannot explicitly locate the position of the solvate THF molecule, i.e., distinguish oxygen atoms among the carbons in THF (type 1 disorder; Figure 4a), since it shows a disorder (type 2) due to the proximity to the two-fold axis. In fact, the residual electron density map indicates a type 2 disorder of the whole molecule over more than two positions; however, only two of them appear as major equivalent positions, which are discussed hereafter. Puckering analysis of the THF [47] was not carried out due to the somewhat low quality of the XRD data and the disorder. We analysed intermolecular distances between atoms of the THF and its neighbouring complexes in the crystal of **1**·THF. Visualization of the THF environment using the Hirshfeld surface technique revealed that one of the non-H atoms of the THF shows contact with the complex slightly shorter than the sum of van der Waals radii (Figure 4b). We assign this atom as the oxygen since otherwise, hydrogens of CH$_2$ moiety of the THF would conflict with the (C)H hydrogen of the complex (the estimated H···H distances of 1.46 Å are unreasonably short). Thus, the THF possesses type 2 disorder (due to the proximity to the two-fold axis), but it does not possess type 1 disorder.

For the lower-symmetry compound **2**·THF, the solvate molecules show less disorder owing to the absence of the special position next to them. One of two crystallographically independent molecules (THF1) is well localized, while another (THF2) shows type 2 disorder. The analysis of intermolecular distances allowed us to unambiguously determine the position of the O atom for both solvate molecules, i.e., to exclude a type 1 disorder, similar to that in **1**·THF. For a comparative analysis of the molecular environment of the THF molecules, we analysed the cavities filled by them. In the absence of THF, structure **1**·THF reveals a void of ca. 160 Å$^3$, which accommodates one THF molecule (Figure 5a). The

calculation with the same parameters for **2**·THF gives coupled voids, which accommodate two crystallographically equivalent molecules, {THF1}$_2$ and {THF2}$_2$ (Figure 5b,c). This suggests that the neighbouring complexes pack in a less dense manner in the case of **2**·THF, leaving a narrow isthmus between the single cavities. With respect to each other, the neighbouring THF1 molecules in structure **2**·THF are arranged in a head-to-head manner (Figure 5e), being closer to each other than in the case of corresponding (closest) positions in **1**·THF (Figure 5d; Table S3). The THF2 molecules arranged in a head-to-tail manner (Figure 5f) reveal approximately the same localization as the correspondingly arranged alternatives in **1**·THF. The tail-to-tail variant, principally implemented in **1**·THF, is not observed in **2**·THF.

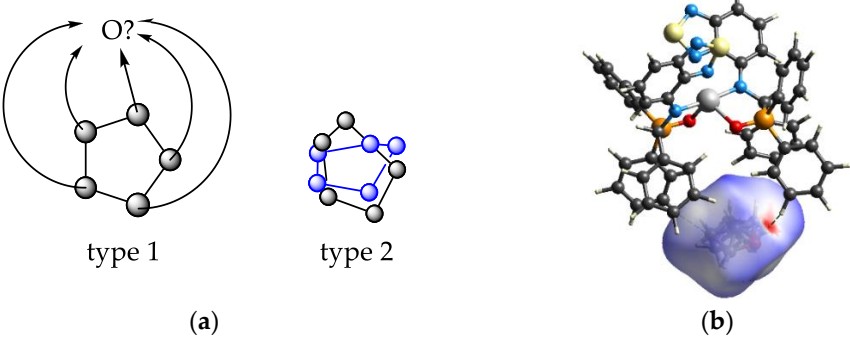

(**a**)            (**b**)

**Figure 4.** (**a**) Schematic representation of the possible types of disorder of THF molecule. (**b**) Hirshfeld surface of a THF molecule in **1**·THF showing the most plausible position of the oxygen and showing the contact (green dashed line) shorter than the sum of van der Waals radii. Regions with distances between atoms less than the sum of their van der Waals radii are shown in red, regions with distances equal to the sum are white, and regions with distances larger than the sum are blue. The surface mapping range: −0.2/1.5 a.u.

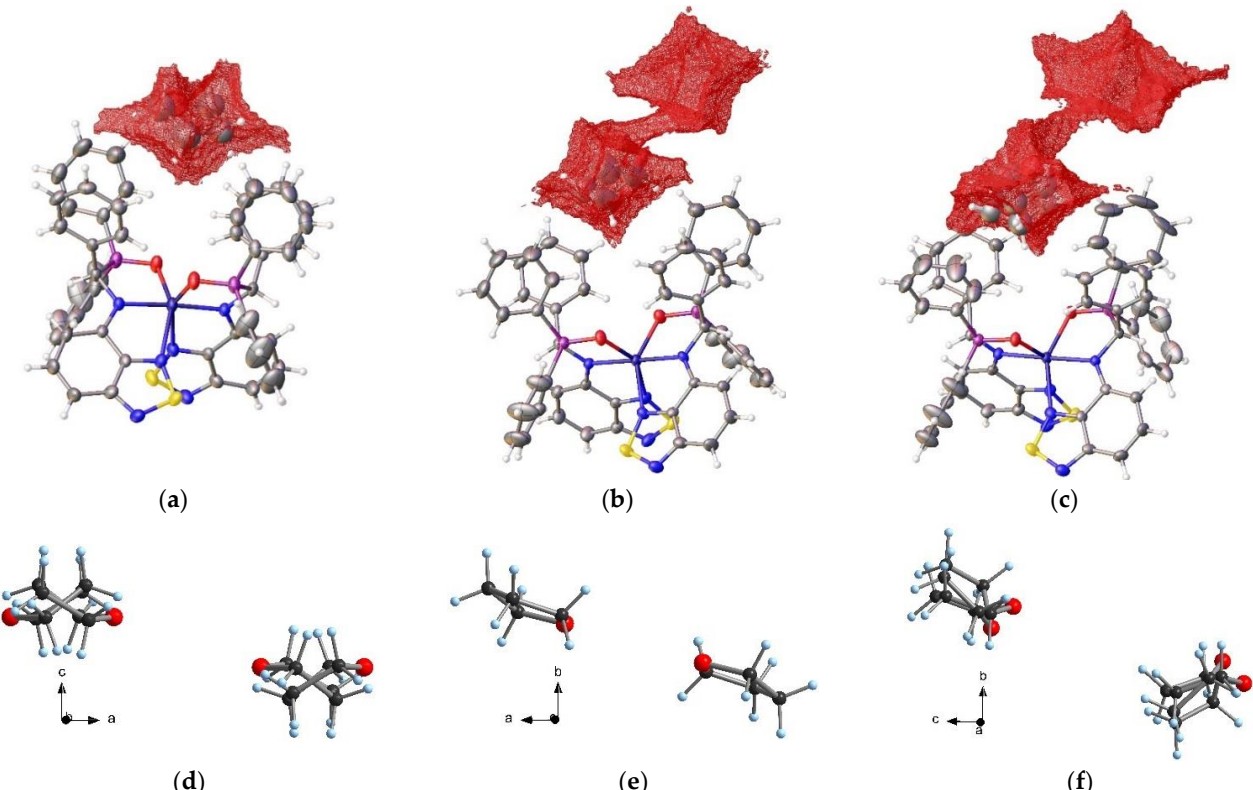

**Figure 5.** Estimated solvent-accessible area in the absence of THF molecules for structures of **1**·THF (**a**) and **2**·THF ((**b**,**c**); for two non-equivalent molecules THF1 and THF2, correspondingly). Relative arrangement of the corresponding THF molecules in **1**·THF (**d**) and **2**·THF (**e**,**f**).

Compound **2**·2THF comprises two crystallographically independent THF molecules; one of them (THF1) is well-ordered to the point that it is reasonably refined in the anisotropic approximation. Another THF molecule (THF2) is poorly located. Analysis of cavities filled by the THF molecules (Figure 6a,b) revealed that, in the absence of the latter, the estimated void volume is 155 Å$^3$ for THF2 in **2**·2THF, which is similar to that in **1**·THF. In the meantime, the cavity occupied by ordered THF1 molecule in **2**·2THF is smaller (130 Å$^3$). This implies that the well localization of the THF1 molecule compared to the others is governed by a lower degree of freedom in its arrangement in the cavity rather than by specific interactions with neighbours. For the THF2 molecule, applying type 2 disorder over two proximate positions gives a reasonable result with a clear residual electron density map (residual peaks are less than $\pm$0.8 $e$). However, no obvious crystal packing conflicts were found for different variants of type 1 positions (Figure S9). Thus, THF2 can exhibit both type 1 and type 2 disorders.

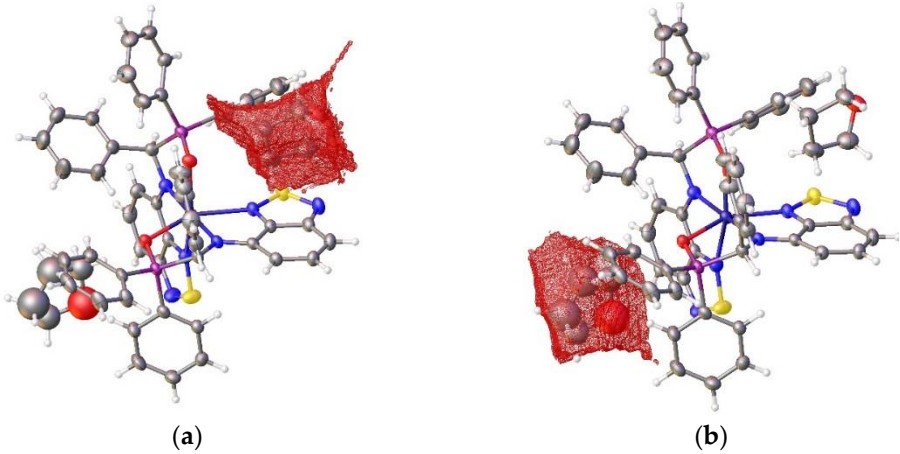

(**a**)　　　　　　　　　　　　　　　　　　　　　　　　　　　　　　(**b**)

**Figure 6.** Estimated solvent-accessible area in the absence of THF1 (**a**) and THF2 (**b**) molecules for the structure of **2**·2THF.

## 4. Conclusions

In summary, we synthesized new closely related zinc and manganese coordination compounds **1–2**. Isotypical compounds **1**·THF and **2**·THF reveal similar crystal packing; the former crystallizes in a high-symmetry space group $I4_1/a$, while the latter features a pseudosymmetrical structure $P2_1/c$ of the doubled volume of the unit cell. The main difference between the structures lies in a different arrangement of solvate THF molecules, which likely modulates the crystal packing of the complexes. More specifically, the arrangement of THF molecules turns from the symmetric disordered case in **1**·THF to the ordered pseudosymmetric one in **2**·THF. Complexes **1** and **2** exhibit quite similar molecular geometry; apparently, the two crystal packings have approximately the same thermodynamic stability, and one or another variant could be obtained under a slight variation of the crystallization conditions. The differences in the conditions, which induce distinctive features of the structures of the compounds, can consist of (1) the different temperatures of the reaction\crystallization (room temperature for M = Zn versus elevated temperature for M = Mn) and (2) some effects of the metal (Zn versus Mn), e.g., seed effects of poorly soluble species [MnCl$_2$(THF)$_x$] (compared to ZnCl$_2$ that is readily soluble). In addition, further treatment of **2**·THF with THF afforded another solvatomorph, **2**·2THF, revealing principally different crystal packing. The reason for implementing the specific structures **1**·THF, **2**·THF, and **2**·2THF is not clear; some minor variations in the reaction/crystallization conditions could influence their formation. The analysis of possible variants for the arrangement of the THF molecules in the crystal structures revealed that in **1**·THF, the molecule possesses a disorder due to the proximity to the two-fold axis. In **2**·THF and **2**·2THF, one crystallographically independent molecule is ordered, while another shows a disorder over two proximate positions. Comparison of the cavities accommodating the solvate

molecules in isotypical **1**·THF and **2**·THF revealed that the neighbouring complexes pack in a less dense manner in the case of **2**·THF, leaving a narrow isthmus between the single cavities. Thus, complexes **1** and **2** featuring well-recognizable molecular geometry are good candidates for further studies of pseudosymmetry, solvate order–disorder relationships, and crystal engineering.

**Supplementary Materials:** The following supporting information can be downloaded at: https://www.mdpi.com/article/10.3390/sym15010157/s1, Figure S1: IR spectra of the compounds; Figures S2–S4: NMR spectra; Table S1: Crystal data and structure refinement for the compounds; Figure S5: Representation of the pseudosymmetry of structure **2**·THF in supergroups; Figure S6: Reciprocal space reconstructions of *h0l* layers for **1**·THF and **2**·THF; Table S2: The Continuous Symmetry Measures analysis and some geometry parameters of the compounds; Figure S7: Overlay of crystal packings of **1**·THF and **2**·THF; Figure S8: A fragment of crystal packing of compound **2**·2THF showing the symmetry operations; Table S3: Estimated volume of voids in the absence of the THF molecules and some geometry characteristics of the relative arrangement of THFs in the compounds; Figure S9: Hirshfeld surfaces of different variants of arrangement of the THF molecule in **2**·2THF.

**Author Contributions:** Conceptualization, T.S.S.; funding acquisition, T.S.S.; investigation, R.M.K. and T.S.S.; supervision, S.N.K.; visualization, R.M.K. and T.S.S.; writing—original draft, T.S.S. and R.M.K.; writing—review and editing, S.N.K. and T.S.S. All authors have read and agreed to the published version of the manuscript.

**Funding:** This research was funded by the Russian Science Foundation (project no. 21-73-10096). The study of the pseudosymmetry of the compounds was supported by the Ministry of Science and Higher Education of the Russian Federation (No. 121031700313-8 and No. 121031700321-3).

**Data Availability Statement:** Data sharing is not applicable to this article.

**Conflicts of Interest:** The authors declare no conflict of interest.

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
