# Peer review of "Synthesis, Crystal Packing Aspects and Pseudosymmetry in Coordination Compounds with a Phosphorylamide Ligand"

_symmetry, doi:10.3390/sym15010157_

Round 1

Reviewer 1 Report

The manuscript by Sukhikh and co-authors report the synthesis and crystal structure of phosphorylmethylamide stabilized complexes of Zn and Mn. The authors in particularly highlight the pseudosymmetry and discuss the role of interactions between solvate and host molecules. It is an interesting work and might be suitable for publication in symmetry. However, authors are requested to please take into consideration the following points while submitting the revised version.

1. In abstract and introduction, n=1,2 only for Mn and not for Zn as Zn always contained one THF as a solvate molecule.

2. Section 2.2, The data collection details for 1 should be incorporated before 2.

3. Section 2.3, KHDMS appears for the first time so it should be explained here instead of in Section 3.1.

4. NMR data of 1 doesn't show any solvated THF molecule, however, elemental analysis does. How is it possible? Could the authors please comment and provide NMR spectra as supoorting information.

5. Authors report doublets at 6.84 and 6.85 ppm, how is it possible?

6. There is no order in presenting the NMR data. For instance 6.82-6.75, 6.35 and then again 6.84 ppm.

7. Authors should report the coupling constants. 

8. Authors have used equimolar amounts of ligand and ZnCl2. Theoritically, the maximum yield can not be more than 50% but authors claim a yield of 95%! Could the authors please explain it?

9. Please provide yield, elemental analysis and NMR data for 2.2THF.

10. Authors attribute the crystal features to either reaction temperature or seeding effects of metal salts. In my opinion, it might be metal ion and crystallization conditions as 2.2THF shows different crystal packing than 2.1THF although the reaction conditions were the same. 

11. Authors should be consistent while using abbreviations. For instance, HL or LH. Preferably, it should be HL.

12. Page 5, lines 163 and 165, what is btd? please be consistent.

13. Characterization data shows one THF solvate molecule for 2.1THF. However, on page 5, line 156 and page 7, authors claim two solvate molecules. Could the authors please explain even though it is crystal packing.

14. Authors should use their full names in manuscript as well as supporting information.

15. There are some minor English and formatting errors which should be corrected.  For instance, in synthetic part, "5 mL THF" should be,"in THF (5 mL)"

Author Response

Merry Christmas, Happy New Year, and thank you very much for the response! We appreciate the comments from the reviewers and provide the point-by-point response below. We have made some changes to the text and the supporting information section, and hope that the improved manuscript will be suitable for publication in its present form.

  1. In abstract and introduction, n=1,2 only for Mn and not for Zn as Zn always contained one THF as a solvate molecule.

We have revised the text specifying n for each metal.

  1. Section 2.2, The data collection details for 1should be incorporated before 2.

We have revised the text by reordering the text.

  1. Section 2.3, KHDMS appears for the first time so it should be explained here instead of in Section 3.1.

We have included the name in Section 3.1.

  1. NMR data of 1 doesn't show any solvated THF molecule, however, elemental analysis does. How is it possible? Could the authors please comment and provide NMR spectra as supoorting information.

We apologize, the corresponding signals for THF were not listed. We have included them in the experimental section as well as included figures of the NMR spectra in the supporting information. For manganese compound 2·THF, the 31P{H} NMR spectrum shows only a weak signal from free HL, which was erroneously attributed to that of the title compound.

  1. Authors report doublets at 6.84 and 6.85 ppm, how is it possible?

This is a typo that has been revised.

  1. There is no order in presenting the NMR data. For instance 6.82-6.75, 6.35 and then again 6.84 ppm.

We have revised the order of the signals.

  1. Authors should report the coupling constants.

We have reported the coupling constants where possible.

  1. Authors have used equimolar amounts of ligand and ZnCl2. Theoritically, the maximum yield can not be more than 50% but authors claim a yield of 95%! Could the authors please explain it?

We apologize for the typo in the loadings (those for ZnCl2 and KHDMS should be swapped); we have revised them.

  1. Please provide yield, elemental analysis and NMR data for 2.2THF.

We grew a few crystals of the compound, which was not further characterized. We have included this information in the experimental section.

  1. Authors attribute the crystal features to either reaction temperature or seeding effects of metal salts. In my opinion, it might be metal ion and crystallization conditions as 2.2THF shows different crystal packing than 2.1THF although the reaction conditions were the same.

Thank you for rising this point. In fact, compounds 1·THF and 2·THF start to precipitate directly from the reaction mixtures, so, technically, the differences are caused by the reaction conditions. However, we agree that the term “crystallization” is more appropriate in this case. Seeding effects of metal salts imply the influence of the metal, although in a narrower sense. We have added a general phrase about the influence of the metal. Since the geometry of complexes 1 and 2 is very similar, and the distortion degree of the latter from the C2 symmetry is low, there are no obvious restrictions for the formation of the tetragonal structure for 2·THF. Note that the structures with related Mn(II) complexes possessing C2 symmetry have been reported elsewhere. Thus, we assume that, in principle, the Mn complex can crystallize in the tetragonal space group, while the Zn in the monoclinic one. The reason for implementing the specific structures 1·THF, 2·THF and 2·2THF is not clear; some minor variations in the reaction/crystallization conditions could influence their formation. We have slightly expanded the corresponding discussion.

  1. Authors should be consistent while using abbreviations. For instance, HL or LH. Preferably, it should be HL.

We have uniformed the abbreviations.

  1. Page 5, lines 163 and 165, what is btd? please be consistent.

We have clarified the “btd” abbreviation (benzothiadiazole)

  1. Characterization data shows one THF solvate molecule for 2.1THF. However, on page 5, line 156 and page 7, authors claim two solvate molecules. Could the authors please explain even though it is crystal packing.

For 2·1THF, there are two crystallographically non-equivalent complexes 2 and two non-equivalent THF molecules (Z’ = 2). We have clarified this in the text.

  1. Authors should use their full names in manuscript as well as supporting information.

We have included the full names.

  1. There are some minor English and formatting errors which should be corrected.  For instance, in synthetic part, "5 mL THF" should be,"in THF (5 mL)"

We have polished the text.

Reviewer 2 Report

1.       I suggest the authors have to highlight the introduction section, its current form is meaningless.

2.       Please provide the IR for the title complex and ligand, and discuss them

3.       Also, give the TGA for confirming the guest molecules.

4.     All the compounds reveal π-π interactions between btd moieties of neighbouring molecules, which possess a similar relative arrangement owing to the similar molecular shape. Some refs could be cited, such as Micropor. Mesopor. Mat, 341(2022) 112098 and Inorganics, 10(2022) 202.

5.      Please discuss the weak interaction of the C..H-O, C..H-N on this system.

6.     I also suggest the author to test the luminesces property of the complex.

7.     The author can exchange the guest solvent by CH3CN or CH3OH?

Author Response

Merry Christmas, Happy New Year, and thank you very much for the response! We appreciate the comments from the reviewers and provide the point-by-point response below. We have made some changes to the text and the supporting information section, and hope that the improved manuscript will be suitable for publication in its present form.

  1. I suggest the authors have to highlight the introduction section, its current form is meaningless.

We do not quite understand the Reviewer; the introduction appears as a separate section that describes the relevance of the topic, introduces some terms used in the text, and references to relevant works (35 references). The short paragraph describes a brief overview of the work. We believe that this is a classic organization of the introduction, and we would like to keep its current form.

  1. Please provide the IR for the title complex and ligand, and discuss them.

We have included the corresponding IR spectra. Since the detailed IR spectroscopy study of the compounds is beyond the aim of the work, we would like not to discuss the spectra.

  1. Also, give the TGA for confirming the guest molecules.

We used THF as a single solvent, so there is no doubt that only THF can act as a solvate molecule in the structures. In addition, single crystal XRD analysis gave reasonable description of the THF solvate molecules, as discussed in detail in Section 3.3. Since the full characterization of the phases is beyond the aim of the current work (the aim is to study crystal packing aspects and pseudosymmetry of the crystals), we would not study them by means of thermogravimetry.

  1. All the compounds reveal π-π interactions between btd moieties of neighbouring molecules, which possess a similar relative arrangement owing to the similar molecular shape. Some refs could be cited, such as Micropor. Mesopor. Mat, 341(2022) 112098 and Inorganics, 10(2022) 202.

The corresponding works on the synthesis and sorption properties of MOFs are interesting. After carefully reading the papers, we decide that they do not include btd derivatives, as well as do not include detailed discussion of π-π interactions. Thus, unfortunately, we cannot link the references with our discussion. However, we slightly expanded the description and cited relevant works.

  1. Please discuss the weak interaction of the C..H-O, C..H-N on this system.

We implicitly analysed intermolecular interactions in section 3.3 when discussed localization of THF molecules. We believe that this is enough for the description of the crystal packing.

  1. I also suggest the author to test the luminesces property of the complex.

Thank you for the suggestion; the study of the luminescent properties of the compounds will be the matter of our future research.

  1. The author can exchange the guest solvent by CH3CN or CH3OH?

Treatment of the compounds with protic agents, e.g. CH3OH would result in hydrolysis of the complexes. Treatment with CH3CN or other aprotic liquids would result in dissolution of the compounds, subsequent crystallization would give structurally different phases. Their study will be the matter of our future research.

Round 2

Reviewer 1 Report

The authors have addressed all the comments and am looking forward to see this nice work in Symmetry.